# Two Heads are Better Than One:
# A Simple Exploration Framework for Efficient Multi-Agent Reinforcement Learning

**Jiahui Li**[1], **Kun Kuang**[1]*, **Baoxiang Wang**[2,3], **Xingchen Li**[1], **Fei Wu**[1], **Jun Xiao**[1], **Long Chen**[4]

[1]Zhejiang University, [2]The Chinese University of Hong Kong, Shenzhen
[3]Shenzhen Institute of Artificial Intelligence and Robotics for Society, [4]HKUST
{jiahuil,kunkuang}@zju.edu.cn, bxiangwang@cuhk.edu.cn,
xingchenl@zju.edu.cn, {wufei,junx}@cs.zju.edu.cn, longchen@ust.hk

## Abstract

Exploration strategy plays an important role in reinforcement learning, especially in sparse-reward tasks. In cooperative multi-agent reinforcement learning (MARL), designing a suitable exploration strategy is much more challenging due to the large state space and the complex interaction among agents. Currently, mainstream exploration methods in MARL either contribute to exploring the unfamiliar states which are large and sparse, or measuring the interaction among agents with high computational costs. We found an interesting phenomenon that different kinds of exploration plays a different role in different MARL scenarios, and choosing a suitable one is often more effective than designing an exquisite algorithm. In this paper, we propose a exploration method that incorporate the CuriOsity-based and INfluence-based exploration (COIN) which is simple but effective in various situations. First, COIN measures the influence of each agent on the other agents based on mutual information theory and designs it as intrinsic rewards which are applied to each individual value function. Moreover, COIN computes the curiosity-based intrinsic rewards via prediction errors which are added to the extrinsic reward. For integrating the two kinds of intrinsic rewards, COIN utilizes a novel framework in which they complement each other and lead to a sufficient and effective exploration on cooperative MARL tasks. We perform extensive experiments on different challenging benchmarks, and results across different scenarios show the superiority of our method.

## 1 Introduction

Multi-Agent Reinforcement Learning (MARL) has been widely used to solve numerous complex real-world problems, such as traffic control [38], autonomous robotics control [17, 26], game AI [25], and network routing [40], *etc.*. Compared with single-agent reinforcement learning, the joint state and action space grows exponentially with the number of agents. To overcome this challenge, the Centralized Training with Decentralized Execution (CTDE) [20, 12] paradigm is proposed, where the joint value function is estimated in the central critic during training, and the local agents execute actions according to their own policies.

Based on the CTDE paradigm, numerous deep MARL methods have been proposed [27, 32, 35, 15]. The most commonly used method for exploration is $\epsilon$-greedy. However, these methods may perform poorly in complex scenarios [13, 29] with a large number of agents and a vast action space. The main issue is that the $\epsilon$-greedy exploration strategy requires a significant number of attempts to become

---

*Correspongding Author

37th Conference on Neural Information Processing Systems (NeurIPS 2023).

familiar with the environment, potentially leading to sub-optimal results. Therefore, an effective exploration strategy is crucial.

Currently, exploration strategies in MARL can be broadly classified into two main categories. The first category includes **curiosity-based exploration** methods [41, 6]. These methods focus on quantifying the unfamiliarity of incoming states and encourage agents to explore a wide range of states that they have not or rarely experienced. The second category comprises **influence-based exploration** methods [9, 33, 39]. These methods aim to promote agents to adopt actions that can influence other agents and achieve better coordination. However, mainstream methods often prioritize the design of sophisticated algorithms within one specific category, which may not perform well in certain scenarios. On the one hand, the joint state space can be excessively large, presenting a challenge for curiosity-based methods to enable agents to experience all unseen states effectively. On the other hand, meaningful interactions among agents can be infrequent, making it difficult for influence-based methods to identify and optimize these interactions. Fortunately, we have discovered that these two types of methods can complement each other, and even a simple combination of them often leads to satisfactory results.

In this paper, we propose an effective and compute-efficient exploration method named COIN, which incorporates CuriOsity-based and INfluence-based exploration. To be specific, we define the influence of one agent on another as the mutual information between its actions and the trajectory of its peer in the next time-step. Our derivation allows us to reformulate the problem of maximizing mutual information into an optimization problem with a tractable lower bound by introducing a variational posterior estimator. This formulation enables us to quantify the impact of each agent's action on the others and use the influence degree as an intrinsic reward to promote exploration. The influence-based intrinsic reward is added to the individual value functions before being sent to the central mixer. Furthermore, this influence-based exploration strategy is compatible with curiosity-based methods, which encourage the agents to explore unfamiliar states. To reduce the exploration state space, we propose an additional method that effectively utilizes the prediction errors of local observations of each agent and the global states to measure the model's uncertainty towards the environment. This uncertainty can be used as an intrinsic reward, which can be added to the extrinsic global reward provided by the environment. Moreover, based on these two types of intrinsic rewards, we have designed a Multi-Agent Reinforcement Learning (MARL) framework where they complement each other, resulting in thorough and efficient exploration without incurring significant computational costs.

Our main contribution can be summarized as follows: (i) We propose a straightforward exploration method based on mutual information that enables the quantification of each agent's influence on the others. (ii) We introduce a simple yet effective approach to measure the model's unfamiliarity with future states, resulting in curiosity-driven exploration. (iii) To achieve effective exploration in MARL settings, we design a framework in which the two kinds of methods complement each other and act as intrinsic rewards which are compute-efficient. (iv) The experiments on three benchmarks StarCraft II [29], MACO [34], and Google Football [13] show the superiority and effectiveness of our COIN.

## 2 Related Work

**Curiosity-based Exploration Methods.** Exploration algorithms focus on encouraging agents to visit a wide range of different states within the environment to avoid getting trapped in sub-optimal results. Among the various branches of exploration methods, curiosity-based approaches [22, 28] have gained significant popularity. One common approach involves designing intrinsic rewards that measure the degree of unfamiliarity of states. Some researchers utilize pseudo-state counts [2, 21, 30] to calculate such rewards, while others [24, 4]. develop models to predict target states and employ the prediction errors as rewards. For instance, Bai et al. [1] propose the utilization of multiple models and compute intrinsic rewards based on the standard deviation of predictions, especially in complex environments. Zheng et al. [41] extends the application of curiosity-based exploration to MARL settings and employs the prediction errors of individual Q-functions as intrinsic rewards. Similarly, Yang et al. [39] designs intrinsic rewards by considering the variance of an ensemble predicting model that incorporates both local observations and global states.

**Influence-based Methods in MARL.** The influence-based exploration strategy is specifically designed for MARL problems, where the focus is on the complex coordination and interactions among

agents. In this context, mutual information serves as an effective and commonly used tool. Jaques et al. [9] introduces the concept of "social influence", represented as $MI(a_i^t; a_j^t | s^t)$, to quantify how an agent's action can influence the action of another agent. This measure captures the interplay and dependencies between agents, providing valuable insights into their collaborative behavior. Similarly,Wang et al. [33] draws inspiration from counterfactual thinking and proposes two exploration techniques: EITI and EDTI. Both of them aim to encourage agents to influence each other's actions. In EITI, mutual information $MI(o_j^{t+1}; o_i^t, a_i^t | o_j^t, a_j^t)$ is also utilized as a formulation to incentivize agents to exert influence on one another, fostering cooperative exploration and coordination.

These influence-based exploration strategies leverage mutual information as a means to capture the interdependencies among agents and promote collaborative exploration in MARL settings. In this paper, we also propose an influence-based intrinsic reward $MI(\tau_j^{t+1}; a_i^t)$ based on mutual information which is simple but efficient. The discussions of the superiority of our method compared with the others are left in Appendix.

**Hybrid Exploration Methods in MARL.** The two exploration strategies mentioned above focus on different aspects, with many studies emphasizing one while neglecting the other. One hybrid method called CIExplore [39] incorporates both strategies by designing them as intrinsic rewards, which is the closest approach to our work. However, CIExplore has shown to be unstable and ineffective in complex MARL scenarios.

In our approach, COIN, we not only ensure compute efficiency in the designed framework for calculating intrinsic rewards, but we also leverage the complementary nature of the two kinds of rewards to achieve sufficient and effective exploration, ultimately enhancing the model capabilities in complex MARL scenarios.

# 3 Preliminaries

**Dec-POMDPs.** The exploration method we discussed in this paper focuses on the settings of decentralized partially observable Markov decision process (Dec-POMDP) [19, 3, 5, 7, 23], which can always be described as a tuple:

$$G = < \mathcal{N}, \mathcal{S}, \mathcal{A}, \mathcal{P}, r, \mathcal{O}, \Omega, \gamma >,$$

where $\mathcal{N}$ represents the set of agents with $|\mathcal{N}| = N$. At each time-step, each agent $i \in \mathcal{N}$ selects and executes an action $a_i \in \mathcal{A}$ to compose a joint action $\boldsymbol{a} \in \mathcal{A}^N$. The environment produces the global state $s \in \mathcal{S}$ according to the transition function $\mathcal{P}(s'|s, \boldsymbol{a}) : \mathcal{S} \times \mathcal{A}^N \to \mathcal{S}$. Meanwhile, the environment provides a global reward shared by all agents according to the function $r(s, \boldsymbol{a}) : \mathcal{S} \times \mathcal{A}^N \to \mathbb{R}$. Moreover, each agent gains the partial observation $o \in \mathcal{O}$ according to the observation function $\Omega(s, i)$ and learns its local policy $\pi^i(a_i|\tau_i) : \mathcal{T} \times \mathcal{A} \to [0, 1]$ conditions on the its local trajectory $\tau_i \in \mathcal{T}$. The goal of all agents is to maximize the discounted cumulative return $\sum_{i=0}^{\infty} \gamma^i r_i$, where $\gamma$ is a discount factor.

# 4 Method

In this section, we introduce two simple yet effective intrinsic rewards for influence and curiosity-based exploration. Following that, we present a framework, as depicted in Fig. 1, that integrates these two strategies to facilitate exploration. Finally, we provide a brief overview of the loss function and offer implementation details, ensuring both effectiveness and computational efficiency in the exploration process.

## 4.1 Influence-based Intrinsic Reward

Compared with single-agent reinforcement learning, complex coordination plays an important role in MARL. As a result, it is essential to incentivize agents to explore states or execute actions that can influence each other. Mutual information [9, 14, 33]is a commonly used tool for quantifying the influence between pairwise agents. Following previous works, we define the influence of the $i$-th agent on the other agents at time-step $\tilde{t}$ as the sum of the mutual information between the trajectory

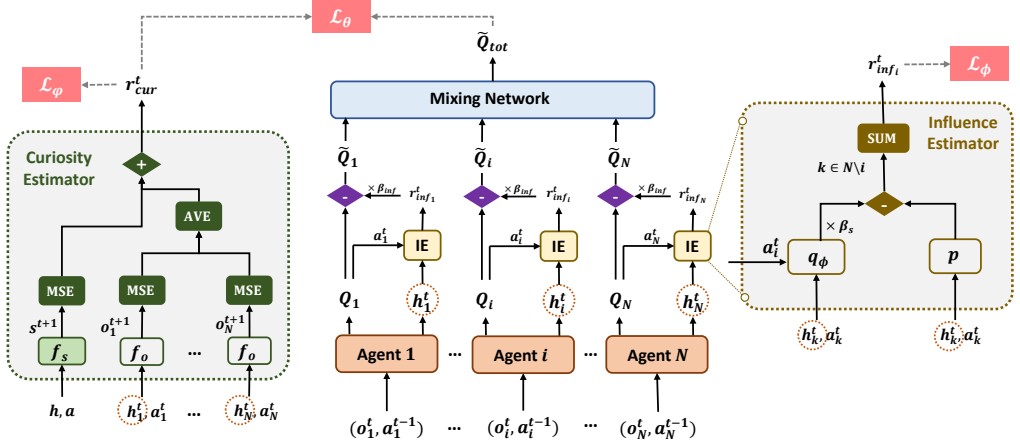

Figure 1: The training framework of COIN. Two additional estimators are set up for computing the intrinsic rewards, where the local observation-action history of all agents are the inputs.

of the other agents and the action taken by the $i$-th agent:

$$\sum_{k \in \mathcal{N} \setminus i} MI(\tau_k^{\tilde{t}+1}; a_i^{\tilde{t}}) = \sum_{k \in \mathcal{N} \setminus i} H(\tau_k^{\tilde{t}+1}) - H(\tau_k^{\tilde{t}+1}|a_i^{\tilde{t}+1}) = \mathbb{E} \sum_{k \in \mathcal{N} \setminus i} [\log \frac{p(\tau_k^{\tilde{t}+1}|a_i^{\tilde{t}})}{p(\tau_k^{\tilde{t}+1})}],$$

where $\tau_k^{\tilde{t}+1}$ represents the trajectory of the $k$-th agent at time-step $\tilde{t} + 1$, and $H(\cdot)$ represents the entropy term.

If we want to promote the influence among agents, we need to maximize $\sum_{k \in \mathcal{N} \setminus i} MI(\tau_k^{\tilde{t}+1}; a_i^{\tilde{t}})$. Meanwhile, we can represent $p(\tau_k^{\tilde{t}+1})$ and $p(\tau_k^{\tilde{t}+1}|a_i^{\tilde{t}})$ as the following equations:

$$p(\tau_k^{\tilde{t}+1}) = p(o_k^0) \prod_{t=0}^{\tilde{t}} p(a_k^t|\tau_k^t)p(o_k^{t+1}|\tau_k^t, a_k^t), \tag{1}$$

$$p(\tau_k^{\tilde{t}+1}|a_i^{\tilde{t}}) = p(o_k^0|a_i^{\tilde{t}}) \prod_{t=0}^{\tilde{t}} p(a_k^t|\tau_k^t, a_i^{\tilde{t}})p(o_k^{t+1}|\tau_k^t, a_k^t, a_i^{\tilde{t}}). \tag{2}$$

Therefore, the mutual information can be rewritten as:

$$\sum_{k \in \mathcal{N} \setminus i} MI(\tau_k^{\tilde{t}+1}; a_i^{\tilde{t}}) = \mathbb{E}[\sum_{k \in \mathcal{N} \setminus i} \{\underbrace{\log \frac{p(o_k^0|a_i^{\tilde{t}})}{p(o_k^0)}}_{\textcircled{1}} + \underbrace{\sum_{t=0}^{\tilde{t}} \log \frac{p(a_k^t|\tau_k^t, a_i^{\tilde{t}})}{p(a_k^t|\tau_k^t)}}_{\textcircled{2}} + \underbrace{\sum_{t=0}^{\tilde{t}} \log \frac{p(o_k^{t+1}|\tau_k^t, a_k^t, a_i^{\tilde{t}})}{p(o_k^{t+1}|\tau_k^t, a_k^t)}}_{\textcircled{3}}\}].$$
$$\tag{3}$$

We can see that Eq. (3) is divided into three terms. Term ① is determined by the environment, and hence can be ignored. Term ② computes the information difference of the $k$-th agent's action selection when the action at time-step $\tilde{t}$ of the $i$-th agent is given. Notice the fact that the action of the $i$-th agent at a particular time can not influence any actions, observations, or trajectories of the other agents at the previous time. Thus, the term ② can also be ignored. Term ③ measures the changes in the target observation of the $k$-th agent. This term can be simplified as $\log \frac{p(o_k^{\tilde{t}+1}|\tau_k^{\tilde{t}}, a_k^{\tilde{t}}, a_i^{\tilde{t}})}{p(o_k^{\tilde{t}+1}|\tau_k^{\tilde{t}}, a_k^{\tilde{t}})}$.

Therefore, we conclude that optimizing the mutual information at Eq. (1) is equivalent to optimize:

$$\mathbb{E}[\sum_{k \in \mathcal{N} \setminus i} \log \frac{p(o_k^{\tilde{t}+1}|\tau_k^{\tilde{t}}, a_k^{\tilde{t}}, a_i^{\tilde{t}})}{p(o_k^{\tilde{t}+1}|\tau_k^{\tilde{t}}, a_k^{\tilde{t}})}].$$

According to the methods of variational inference [31, 14], we can optimize a tractable evidence lower bound of the above expectation by introducing a variational posterior estimator. Hence, we have the following equation:

$$\mathbb{E}[\log \frac{p(o_k^{\tilde{t}+1}|\tau_k^{\tilde{t}}, a_k^{\tilde{t}}, a_i^{\tilde{t}})}{p(o_k^{\tilde{t}+1}|\tau_k^{\tilde{t}}, a_k^{\tilde{t}})}] \geq \mathbb{E}[\log \frac{q_\phi(o_k^{\tilde{t}+1}|\tau_k^{\tilde{t}}, a_k^{\tilde{t}}, a_i^{\tilde{t}})}{p(o_k^{\tilde{t}+1}|\tau_k^{\tilde{t}}, a_k^{\tilde{t}})}], \tag{4}$$

where $q_\phi$ represents the variational posterior estimator parameterized by $\phi$, and we omit the summation notation for convenience.

For encouraging the interaction among agents and promoting exploration, we design an influence-based intrinsic reward for each agent. The intrinsic reward for agent $i$ at time-step $\tilde{t}$ is computed as:

$$r_{inf_i}^{\tilde{t}} = \sum_{k \in \mathcal{N} \setminus i} [\beta_s \log q_\phi(o_k^{\tilde{t}+1}|\tau_k^{\tilde{t}}, a_k^{\tilde{t}}, a_i^{\tilde{t}}) - \log p(o_k^{\tilde{t}+1}|\tau_k^{\tilde{t}}, a_k^{\tilde{t}})], \tag{5}$$

where $\beta_s$ is a scaling factor to guarantee $r_{inf_i}^{\tilde{t}} > 0$.

## 4.2   Curiosity-based Intrinsic Reward

Curiosity-based exploration is a widely adopted approach in single-agent reinforcement learning, with the objective of encouraging agents to explore unfamiliar states. Typically, this is achieved by computing intrinsic rewards based on prediction errors [24, 4] or variances [1] of future states, rewards, or value functions. In the context of MARL, curiosity-based exploration remains essential. However, the prediction of joint states becomes unreliable due to the exponential increase in state space with the number of agents, and the scarcity of meaningful states. Fortunately, the local observations of each agent can serve as a measure of curiosity. For simplicity, we use the prediction error of both the global state and the local observations to calculate the curiosity-based intrinsic reward in the partially observable environment as:

$$r_{cur}^{\tilde{t}} = \frac{1}{|\mathcal{N}|} \sum_{i=1}^{\mathcal{N}} (o_i^{\tilde{t}+1} - f_{o_i}(\boldsymbol{\tau}^{\tilde{t}}, \boldsymbol{a}^{\tilde{t}}))^2 + (s^{\tilde{t}+1} - f_s(\boldsymbol{\tau}^{\tilde{t}}, \boldsymbol{a}^{\tilde{t}}))^2, \tag{6}$$

where $f_{o_i}(\boldsymbol{\tau}^{\tilde{t}}, \boldsymbol{a}^{\tilde{t}})$ and $f_s(\boldsymbol{\tau}^{\tilde{t}}, \boldsymbol{a}^{\tilde{t}})$ represent the predicting network to estimate the observation of each agent and global state at time-step $\tilde{t} + 1$, and $\boldsymbol{a}^{\tilde{t}}, \boldsymbol{\tau}^{\tilde{t}}$ represent the joint action, and trajectories of all agents at time-step $\tilde{t}$.

## 4.3   Implementation of the Estimators

Our goal is to develop two effective estimators for computing intrinsic rewards as well as make them compute-efficient. For the influence estimator, we model $q_\phi(o_k^{\tilde{t}+1}|\tau_k^{\tilde{t}}, a_k^{\tilde{t}}, a_i^{\tilde{t}})$ by a variational posterior estimator, which is updated via loss:

$$L_\phi = \sum_{i \in \mathcal{N}} \sum_{k \in \mathcal{N} \setminus i} (o_k^{\tilde{t}+1} - q_\phi(o_k^{\tilde{t}+1}|\tau_k^{\tilde{t}}, a_k^{\tilde{t}}, a_i^{\tilde{t}}))^2. \tag{7}$$

Meanwhile, the $p(o_k^{\tilde{t}+1}|\tau_k^{\tilde{t}}, a_k^{\tilde{t}})$ is approximated by a prediction network $p_\eta$ updated via loss:

$$L_\eta = \sum_{i \in \mathcal{N}} \sum_{k \in \mathcal{N} \setminus i} (o_k^{\tilde{t}+1} - p_\eta(o_k^{\tilde{t}+1}|\tau_k^{\tilde{t}}, a_k^{\tilde{t}}))^2. \tag{8}$$

For the curiosity estimator with parameters $\varphi$, we update it via loss:

$$L_\varphi = \frac{1}{|\mathcal{N}|} \sum_{i=1}^{\mathcal{N}} (o_i^{\tilde{t}+1} - f_{o_i}(\boldsymbol{\tau}^{\tilde{t}}, \boldsymbol{a}^{\tilde{t}}))^2 + (s^{\tilde{t}+1} - f_s(\boldsymbol{\tau}^{\tilde{t}}, \boldsymbol{a}^{\tilde{t}}))^2. \tag{9}$$

Notice that the summation symbol can become an obstacle when computing intrinsic rewards and losses efficiently. In order to accelerate the training procedure, we have implemented the two estimators in a compute-efficient manner, as illustrated in Fig. 2.

In the influence estimator, Eq. (7) requires us to propagate $q_\phi$ for $N-1$ iterations in order to obtain the intrinsic reward for a single agent. To improve efficiency, we have designed a batch implementation for $\sum_{k \in \mathcal{N} \setminus i}$, allowing us to estimate the observations of all other agents simultaneously in a single forward propagation. Likewise, $p(\cdot | \tau_k, a_k)$ also adopts the same trick.

As for the curiosity estimator, all $f_{o_i}$ have the same inputs. To streamline this process, we implement them using a single neural network that outputs a concatenated tensor.

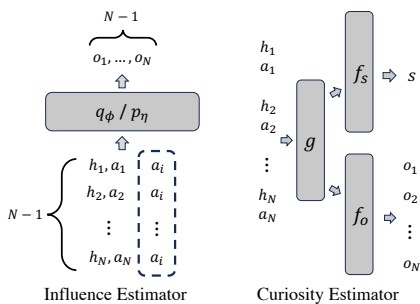

Figure 2: Implementations of the two estimators for computing the summation symbol in a single forward propagation.

To further alleviate the computational burden, we adopt the parameter-sharing technique, and all agents share the same network in the influence estimator. Similarly, for curiosity-based intrinsic reward, $f_s$ and $f_o$ share the same encoder $g$ to get the representations. Inspired by the approach presented in [15], we represent the trajectory of the agent using the hidden state of a recurrent neural network. Consequently, we have $h_i^{\tilde{t}} = \tau_i^{\tilde{t}}$ and $\boldsymbol{h}^{\tilde{t}} = \boldsymbol{\tau}^{\tilde{t}}$. Meanwhile, for the learning more stable, we compute the intrinsic rewards in Eq. (5) and Eq. (6) via target networks $\phi'$, $\eta'$ and $\varphi'$ which are copied from $\phi$, $\eta$ and $\varphi$ periodically.

## 4.4 Incorporation of the Two Heads

We would like to emphasize the effectiveness of combining these two methods in a straightforward manner, rather than designing a intricate algorithm that solely focuses on one aspect. There are two reasons for this: (i) Curiosity-based intrinsic rewards are designed to encourage exploration of unfamiliar states, while influence-based intrinsic rewards aim to promote coordination among agents. These rewards operate on different dimensions and play distinct roles in various scenarios. (ii) Either of the methods may not work in certain training time-steps, as no method can guarantee constant positive influence among agents, and the meaningful unfamiliar joint state is too sparse. Some recent works [33, 39] attempt to use both kinds of methods, however, concerns regarding their effectiveness and computational efficiency remain which lead the unsatisfactory results in complicated scenarios.

In this subsection, we will introduce how to incorporate them under the paradigm of CTDE. For simplicity, we omit the superscript of time-step in this subsection. For the influence-based intrinsic rewards, we apply it to the individual local value functions of each agent to promote the interaction. For $i$-th agent it is computed as:

$$\tilde{Q}_i = Q_i - \beta_{inf} * r_{inf_i}, \tag{10}$$

where $Q_i$ is the local value function, and $\beta_{inf}$ is the scale factor. $\tilde{Q}_i$ will then be sent to the central critic $Mixer$ to get the joint value function:

$$\tilde{Q_{tot}} = Mixer(\tilde{Q}_1, ..., \tilde{Q}_N). \tag{11}$$

Notice that the intrinsic rewards are incorporated before the value functions are sent to the central mixer. This is done to ensure that the intrinsic rewards can be scaled by the network appropriately. Meanwhile, it is worth mentioning that the intrinsic rewards are not applied to the target network. The reasons for this decision will be discussed in the Appendix.

The curiosity-based intrinsic rewards are directly added to the extrinsic rewards, and subsequently, the agents and the central mixer are updated using TD-loss:

$$L_\theta = (y - \tilde{Q_{tot}}), \tag{12}$$

$$y = r + \beta_{cur} * r_{cur} + \gamma * Q'_{tot}, \tag{13}$$

where $Q'_{tot}$ represents the target network, $\beta_{cur}$ represents the scale factor.

We show the algorithm of our COIN in *Alogrithm* 1.

**Algorithm 1** Influence and Curiosity-based Exploration

**Initialize:** Local agents and central critic $\theta$ with its target networks $\theta'$, max episode length $T$, networks of the estimator for computing intrinsic rewards $\phi$, $\eta$ and $\varphi$ with its target networks $\phi'$, $\eta'$ and $\varphi'$.

1: **for** each training episode **do**
2:     **while** global state $s \neq$ terminal **and** time step $t < T$ **do**
3:         $t = t + 1$
4:         **for** each agent $i$ **do**
5:             Compute the local value function $Q_i$ and get the hidden state $h_t^i$
6:             Select action $a_t^i$ and execute
7:         **end for**
8:         Execute the joint action $\boldsymbol{a} = (a_t^1, a_t^2, ..., a_t^n)$
9:         Get reward $r_{t+1}$ and next state $s_{t+1}$
10:     **end while**
11:     Add episode to replay buffer
12:     Collate episodes in the buffer into a single batch
13:     **for** $t = 1$ to $T$ **do**
14:         Compute the influence-based intrinsic reward via Eq. (5)
15:         Compute the Curiosity-based intrinsic reward via Eq. (6)
16:         Add the influence-based to individual value functions via Eq. (10)
17:         Compute the joint value function via Eq. (5)
18:         Compute the targets $y$ via Eq. (13)
19:         Update $\theta$ by minimizing the TD-loss in Eq. (12)
20:         Update $\phi$, $\eta$ and $\varphi$ by minimizing the loss in Eq. (7),Eq. (8) and Eq. (9)
21:         Update $\theta' = \theta$, $\phi' = \phi$, $\eta' = \eta$, $\varphi' = \varphi$ periodically
22:     **end for**
23: **end for**

## 5 Experiments

Our experiments aim to answer the following questions: (1) Q1: Is the exploration strategy important in MARL? (2) Q2: How does each kind of exploration strategy affect the model performance in different scenarios? (3) Q3: Can influence-based exploration cooperate with curiosity-based exploration?

We performed experiments on three popular MARL benchmarks: StarCraft II micro management challenge (SMAC)[29], MACO[36], and Google Research Football (GRF) [13], with different scenarios. The details of each benchmark will be introduced in the Appendix. We adopted two popular baselines, QMIX [27] and QPLEX [32], and applied our exploration method to them. To further demonstrate the effectiveness of our exploration method, we chose the curiosity-based exploration method EMC [41], the influence-based exploration method EITI [33], and the hybrid method CIExplore [39] as strong baselines. It should be mentioned that EITI and CIExplore are implemented based on the actor-critic framework in the original paper. For a fair comparison, we implemented EITI and CIExplore based on QMIX under the same framework as us. Meanwhile, since we aim to verify the effectiveness of different exploration strategies, we do not use the special episode buffer in EMC, as well as the other tricks in all methods. All the curves in the figures of this section are the mean of 10 random seeds, which have been smoothed, and the shaded areas represent the standard deviation. The experiment settings are also provided in the Appendix.

### 5.1 Performance on several benchmarks (Q1)

**Performance on StarCraft II benchmark.** We first conduct experiments on the most popular environment, StarCraft II micro management challenge. StarCraft II is a real-time strategy game. We set the difficulty of game AI at the "very difficult" level. This benchmark consists of various maps with different levels of challenges, "easy", 'hard', and "super hard". We select three representative

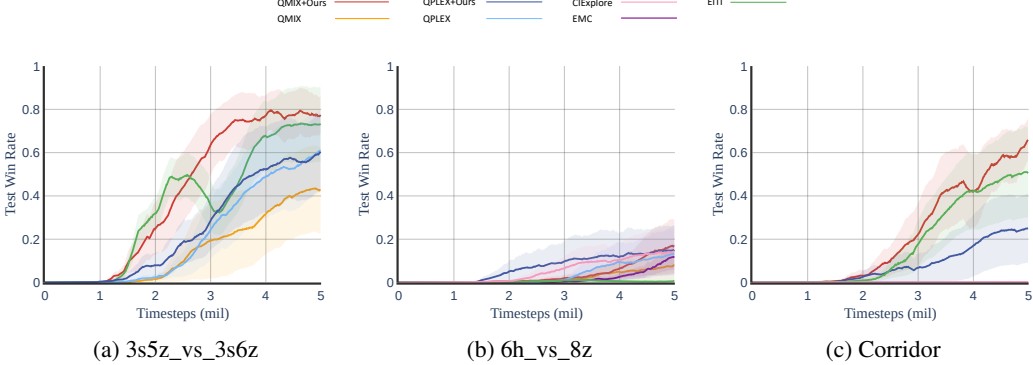

Figure 3: Performance of our exploration method on three scenarios in StarCraft II benchmark.

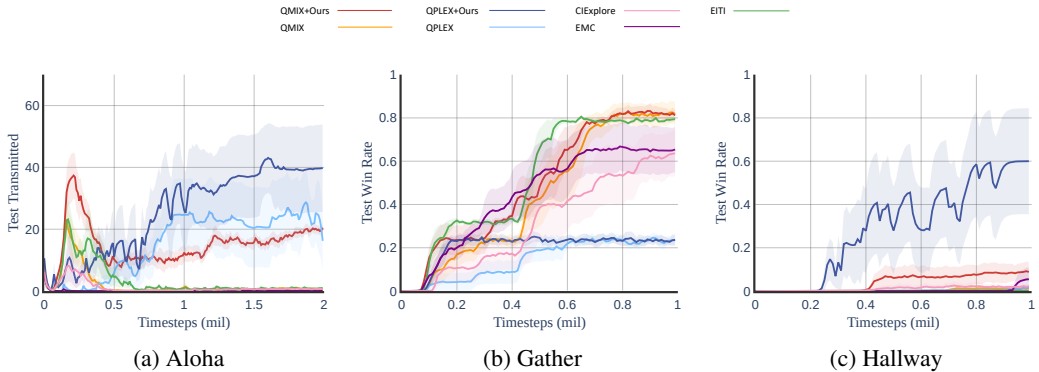

Figure 4: Performance of our exploration method on three scenarios in MACO benchmark.

and super hard scenarios, *3s5z_vs_3s6z*, *corridor*, and *6h_vs_8z* to carry out experiments[2], and the results are depicted in Fig. 3. Our method shows a significant improvement over the baselines in all maps, highlighting the importance of the exploration strategy in MARL. Furthermore, the superior results compared to other exploration baselines demonstrate the effectiveness of our COIN method. We also conducted experiments with sparse reward setting, and the results are provided in the Appendix.

**Performance on MACO benchmark.** Next, we evaluated our method in the MACO benchmark, which features relatively sparse rewards. MACO incorporates several coordination tasks from various multi-agent learning literature and enhances their difficulty. We conducted experiments on three MACO scenarios: *Aloha*, *Hallway*, and *Gather*. These scenarios require different levels of agent coordination. The results of our experiments are presented in Fig. 4.

In the Aloha scenario [8, 18], the agents strive to achieve as many successful transmissions as possible within a limited number of time steps. None of the other exploration methods proved effective in this scenario. Our COIN approach improved the two baselines by nearly 20 mean transmissions. The results of all methods were peculiar, with an initial burst followed by a drop and then plateaued poor results. Initially, the agents attempted exploration, resulting in more successful transmissions. However, the punishment mechanism in this setting was overly severe, causing the agents to adopt conservative strategies. In the *Gather* scenario, the task is relatively simple, with agents aiming to choose the same action to achieve common goals set at the beginning of each episode. Although exploration did not significantly improve the performance of the baselines, our COIN approach still achieved a faster convergence speed. There was a failure case known as the "noisy TV" in the Gather scenario. Choosing the goal set at the beginning of the episode resulted in a higher global reward, while not choosing this goal led to a relatively lower reward. It is evident that QPLEX failed

---

[2]We do not carry out the experiment of EMC on corridor because it takes too much computational resources which is out of the max memory of our GPU (GTX 2080 Ti).

and produced suboptimal results, and our framework did not provide significant improvement. The *Hallway* scenario [34] involved agents divided into several groups, where agents within the same group aimed to reach the same state simultaneously, while different groups aimed to reach different states. Our COIN approach succeeded with QPLEX, resulting in a 60% improvement in mean win rates. However, applying any exploration method to QMIX showed little improvement, which can be considered another failure case.

**Performance on GRF benchmark.** In the GRF game, players need to coordinate with each other to organize offense and score goals. The rewards in GRF are sparse, with agents only receiving rewards at the end of the game. Therefore, exploration plays a vital role in guiding the agents to discover winning strategies.

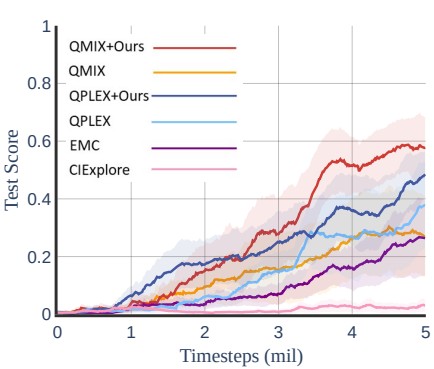

Figure 5: Experiment results on GRF benchmark in *academy 3 vs 1 with keeper*.

We compared the performance of several baselines on the GRF benchmark, as shown in Fig. 5. The curve representing our COIN approach demonstrates the application of our exploration framework on QMIX. Our approach outperforms in the scenario "academy 3 vs 1 with keeper". QMIX and QPLEX fail to perform well due to the sparse rewards. EMC applies a curiosity-based exploration strategy to QPLEX but results in a decrease in model performance. In contrast, our COIN approach combines influence-based exploration and curiosity-based exploration, outperforming all baselines with an improvement of nearly 0.2 mean scores per episode.

## 5.2 Ablations and Interesting Findings (Q2, Q3)

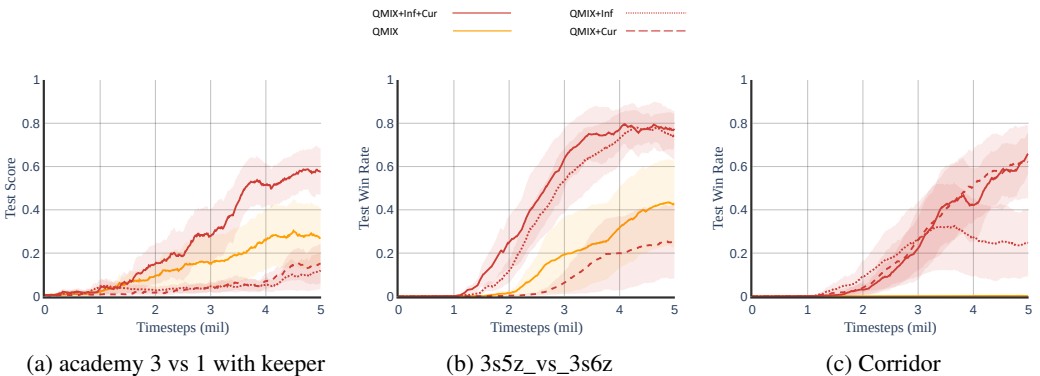

(a) academy 3 vs 1 with keeper     (b) 3s5z_vs_3s6z     (c) Corridor

Figure 6: Ablation study of our COIN on GRF and SMAC benchmark.

To understand how influence-based exploration and curiosity-based exploration work in different scenarios, we conducted ablation studies. We selected three representative scenarios to showcase these results, as depicted in Fig. 6. The dotted line represents applying only the influence-based intrinsic reward on QMIX, while the dashed line represents applying only the curiosity-based intrinsic reward. Interestingly, we found that different exploration strategies play different roles in different scenarios. When only influence-based or curiosity-based exploration was conducted in the *academy 3 vs 1 with keeper* scenario, the model did not benefit from the exploration strategy. Instead, the performance decreased by nearly 0.5 mean scores. To understand this finding, we consulted literature and watched replays, and find that this phenomenon arises from the parameter sharing technique. According to Li et al. [14], shared parameters among agents can lead to similar behaviors, hindering the model's ability to learn successful policies in challenging tasks. Our influence-based intrinsic reward promotes agents to influence each other, bringing diversity to their behaviors. However, focusing solely on influence

among agents neglects crucial patterns, such as cutting to attract the enemy defender, resulting in poor outcomes. Fortunately, our proposed method succeeded by striking a balance between influence and curiosity. This interesting result also explains why EMC decreases the baseline performance in this scenario, as shown in Fig. 5. In the scenario *3s5z_vs_3s6z*, influence-based exploration plays a significant role, while curiosity-based exploration harms the performance. This is because the learned winning strategy involves one of the allied units kiting several enemies to the corner of the map and sacrificing itself, thus not requiring much exposure to unknown states. In the *corridor* scenario, both exploration strategies improve the baseline performance significantly. However, curiosity-based exploration achieves even greater improvement and reaches the best performance.

In all scenarios, the best performance is achieved when both kinds of exploration strategies are incorporated. Therefore, we conclude that different scenarios require different exploration strategies, and when we are unsure which one will be effective, incorporating both is often a good choice. For more experimental results, please refer to the Appendix.

### 5.3 The Chosen hyperparameters

In Section 4, we introduced several hyperparameters for computing intrinsic rewards. To ensure stability and guarantee that the influence-based intrinsic reward is larger than zero, we set $\beta_s$ in Eq. (5) to 0.5 for all scenarios in all benchmarks. Additionally, $\beta_{inf}$ and $\beta_{cur}$ are scaling factors used to adapt the two types of intrinsic rewards to the environment. Given that the importance of the two types of exploration may vary at different time steps, determining which one will dominate is challenging. To address this, we combine the two types of intrinsic rewards in a straightforward manner. When designing the hyperparameters, we follow two principles: (i) ensuring that both types of intrinsic rewards are on the same order of magnitude, and (ii) scaling the intrinsic rewards to be approximately 100 to 1000 times lower than the maximum extrinsic reward at the beginning of the training stage. Details of all the hyperparameters are listed in Appendix.

## 6 Conclusion

In this paper, we explore an interesting phenomenon where different types of exploration play varying roles in different scenarios of MARL. However, most current studies tend to focus on one type of exploration and overlook the other. In light of this, we propose a framework called COIN that effectively combines and complements these two types of exploration strategies. Firstly, COIN measures the influence of each agent on the others using mutual information theory, and assigns it as intrinsic rewards to individual value functions. Additionally, COIN calculates curiosity-based intrinsic rewards by utilizing prediction errors, which are then added to the extrinsic reward. By integrating these two types of intrinsic rewards, COIN creates a novel framework where they complement each other, ultimately leading to more efficient and effective exploration in cooperative MARL tasks. Both of these types of intrinsic rewards are computationally efficient within our designed framework, thus enabling effective exploration. Our experiment results on various benchmarks demonstrate the superiority of our proposed method.

**Limitations and future work.** Though the incorporation of two kinds of exploration is useful, our method relies on scaling hyperparameters and cannot automatically weigh the two kinds of strategies. In future work, we will investigate how to discover the causal relations among agents. With these potential interdependencies, we can assign appropriate weights to each agent and avoid computing the MI for unnecessary agent pairs.

## Acknowledgements

This work was supported by the National Key Research & Development Project of China (2021ZD0110700), the National Natural Science Foundation of China (62006207, 62037001, U19B2043, 61976185), the StarryNight Science Fund of Zhejiang University Shanghai Institute for Advanced Study (SN-ZJU-SIAS-0010), and the Fundamental Research Funds for the Central Universities (226-2023-00048, 226-2022-00051, 226-2022-00142). Baoxiang Wang is partially supported by The National Natural Science Foundation of China (62106213 and 72150002) and Shenzhen Science and Technology Program (RCBS20210609104356063 and JCYJ20210324120011032).

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

# A  Appendix

## A.1  Details of the benchmarks

**Multi-agent coordination challenge benchmark.**    Wang et al. [36] collect six coordination tasks from various multi-agent learning literature to create the Multi-Agent Coordination Challenge (MACO) benchmark. For our experiments, we have selected three tasks from this benchmark.

The *Aloha* coordination task [8, 18] consists of a $2 \times 5$ array with 10 agents. The objective is to send messages and maximize the backlog, which has a maximum value of 5. However, if two adjacent agents send messages simultaneously, a collision occurs, and the messages need to be resent. For each successful transmission, a positive global reward of 0.1 is gained, while a collision results in a negative global reward of -10. Each unit initially has 1 backlog, and at each time step, there is a probability of 0.6 for a new packet to arrive, provided that the maximum backlog has not been reached yet. In *Gather* [37], there are 5 agents whose objective is to collectively choose the same action in order to achieve common goals. These goals are established at the beginning of each episode. If all agents successfully choose the goal, the entire team receives a reward of 10. On the other hand, if none of the agents chooses the goal, a reward of 5 is given to the team. *Hallway* [34] involves 12 agents organized into multiple groups. Each agent is randomly assigned to a state within each chain. In this task, if agents belonging to the same group reach the same state at the same time, they receive a global reward. However, if more than one group reaches the same state, a global punishment is given.

**StarCraft II micro management benchmark.**    The StarCraft II micromanagement challenge [29] involves agents controlling allied armies and collaborating to engage with enemy forces controlled by the game engine. This benchmark includes different maps that pose varying levels of challenges, categorized as "easy", "hard" and "super hard". For our experiments, we have chosen three representative super hard scenarios: *3s5z_vs_3s6z*, *corridor*, and *6h_vs_8z*. The game AI difficulty is set to "very difficult" level. Table 1 provides an introduction to these selected maps.

Table 1: Experiment maps of StarCraft II micro management benchmark.

| Map Name | Ally Units | Eenemy Units | Agent Type | Difficulty |
|---|---|---|---|---|
| 3s5z_vs_3s6z | 3 Stalkers, 5 Zealots | 3 Stalkers, 6 Zealots | Heterogeneous | Super hard |
| 6h_vs_8z | 6 Hydraliks | 8 Zealots | Homogeneous | Super hard |
| corridor | 6 Zealots | 24 Zerglings | Homogeneous | Super hard |

**Google research football benchmark.**    In the Google Research Football (GRF) game [13], players are required to coordinate with each other to organize offensive plays and score goals. In this game, agents control the players on the left side, excluding the goalkeeper, while the players on the right side are controlled by rule-based bots using hand-coded built-in AI. Each player can observe the positions and movement directions of all players, as well as the football. They have a discrete action space that includes options for moving, sliding, shooting, and passing. Soccer players must determine the optimal timing for shooting by combining dribbling and passing techniques. In this game environment, rewards are only provided at the end of the game, with a reward of 100 for scoring a goal and a penalty of -1 for losing the game.

## A.2  Experiment settings

We list the experiment settings and hyperparameters in Table 2. The two important scale factors $\beta_{cur}$ and $\beta_{inf}$ in MACO depends on the scenarios. We set $\beta_{cur} = \beta_{inf} = 0.05$ in *aloha*, $\beta_{cur} = \beta_{inf} = 0.01$ in *gather*, and $\beta_{cur} = \beta_{inf} = 0.0005$ in *hallway*.

## A.3  Comparison with the other exploration methods

In this paper, we employ both influence-based and curiosity-based exploration strategies. In comparison to other methods, we utilize a simple yet effective approach to implement these two types of exploration. In this subsection, we will discuss the differences among these methods.

Table 2: Hyperparameters and experiment settings

| Settings | StarCraft II | MACO | GRF |
|---|---|---|---|
| Batch size | 32 | 32 | 32 |
| Training episodes | 5mil | 1mil(2mil for *aloha*) | 5mil |
| Agent optimiser | RMSProp | RMSProp | Adam |
| Estimator optimiser | Adam | Adam | Adam |
| Agent learning rate | 0.0005 | 0.0005 | 0.0005 |
| Estimator learning rate | 0.0005 | 0.0005 | 0.0005 |
| Target central critic update interval | 200 episodes | 200 episodes | 200 episodes |
| $\beta_{cur}$ | 0.0005 | 0.05/0.01/0.0005 | 0.005 |
| $\beta_{inf}$ | 0.0005 | 0.05/0.01/0.0005 | 0.005 |
| $\beta$ | 0.5 | 0.5 | 0.5 |

**Mutual information and influence-based exploration.** Mutual information is a commonly utilized concept in existing works for multi-agent reinforcement learning (MARL) problems. MOA [9] was the first to propose using $MI(a_i^t; a_j^t|s^t)$ as intrinsic rewards. Another formulation called EITI [33] is given by $MI(o_j^{t+1}; o_i^t, a_i^t|o_j^t, a_j^t)$. Similarly, PMIC [16] proposes $MI(a; s)$ as a collaboration criterion. Additionally, Jiang and Lu [11] propose $MI(I; O)$, while Li et al. [14] propose $MI(\tau; I)$, where $I$ represents the agent's ID. The main differences among these methods lie in their intentions. The first three methods aim to promote influence and collaboration among agents, while the last two aim to enhance diversity among the agents.

In this paper, we also incorporate mutual information $MI(\tau_j^{t+1}; a_i^t)$ to quantify the influence of one agent's action on another agent. While the basic motivation aligns with MOA and EITI, our proposed approach is (a) more reasonable and (b) easier to compute. The reasons for these distinctions are as follows: **(a) More reasonable:** Intuitively, an agent can only influence the future behavior of other agents. Therefore, we consider $MI(a_i^t; a_j^t|s^t)$ as less reasonable. Additionally, the crucial factor that influences other agents is the action taken, rather than the current states or observations. Consequently, $MI(\tau_j^{t+1}; a_i^t)$ is more reasonable than $MI(o_j^{t+1}; o_i^t, a_i^t|o_j^t, o_j^t)$.

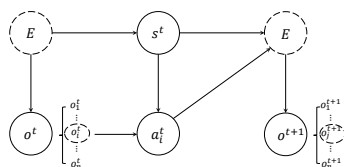

Figure 7: Markov Decision Process in MARL.

Specifically, in the computation of the $i$-th influence on $j$, $MI(\tau_j^{t+1}; a_i^t) = \log \frac{p(\tau_j^{t+1}|a_i^t)}{p(\tau_j^{t+1})}$, whereas $MI(o_j^{t+1}; o_i^t, a_i^t|o_j^t, a_j^t) = \log \frac{p(o_j^{t+1}|o_i^t, a_i^t, o_j^t, a_j^t)}{p(o_j^{t+1}|o_j^t, a_j^t)}$. Referencing the Markov decision process in Fig. 7, in $p(o_j^{t+1}|o_i^t, a_i^t, o_j^t, a_j^t)$, $o_i^t$ becomes the noise factor in predicting $o_j^{t+1}$, which can potentially hinder accurate influence estimation in certain scenarios. For instance, let the optimal policy $\pi(a_i^t|o_i^t, s_i^t)$ be an *and* operation, and let the observation function $g(o_j^t|\cdot)$ be an element-wise *xor* operation for all inputs. $o_j^{t+1}$ should be 0 when $o_i^t = 1$ and $s^t = 0$, but introducing $o_i^t = 1$ into $g$ would result in $o_j^{t+1} = 1$. **(b) Easier to compute:** We have designed a framework where the intrinsic reward at each time step can be computed in a single forward propagation. This approach is straightforward to implement and does not impose a significant computational burden.

**Prediction network and curiosity-based exploration.** Curiosity-based exploration methods, widely employed in reinforcement learning [22, 28], typically involve designing intrinsic rewards to measure the degree of unfamiliarity associated with states. A common approach entails setting up one or several networks to predict information about the next time-step, with the instability of these predictions serving as the intrinsic reward. For instance, Pathak et al. [24] and Burda et al. [4] consider the prediction error of target states as intrinsic rewards. Bai et al. [1] use the standard deviation of target Q-function predictions as the intrinsic reward. Zheng et al. [41] utilize the prediction errors of individual Q-functions as intrinsic rewards in multi-agent reinforcement learning (MARL) problems. Similarly, Yang et al. [39] design intrinsic rewards based on the variance of an ensemble predictive model incorporating local observations and global states. In our approach, we adopt a similar strategy to the aforementioned methods. We use the prediction errors of next

states and observations as intrinsic rewards. Since the observations and global states are provided by the environment, we consider them to be more reliable supervision information compared to value functions. Moreover, these prediction errors can capture the unfamiliarity from two perspectives: local observations for individual unfamiliarity and global states for joint unfamiliarity. Consequently, this approach is straightforward yet effective in MARL settings.

## A.4 Two heads are better than one

One key highlight of this paper is the effectiveness of incorporating both methods rather than solely focusing on a single aspect through an elegant algorithm. To support our viewpoint, we evaluate different exploration strategies on the baselines [27, 32] and present the results on the MACO benchmark in Fig. 8 amd Fig 9. In the figure, the dashed line represents solely applying the curiosity-based intrinsic reward, while the dotted line represents solely applying the influence-based intrinsic reward. It is worth noting that either exploration strategy may fail in certain scenarios. However, by incorporating both methods, we consistently achieve the best results. We also perform experiments on the StarCraft II benchmark with sparse reward setting, and the results can is show in Fig. 10, in which MASER [10] is a goal based exploration method.

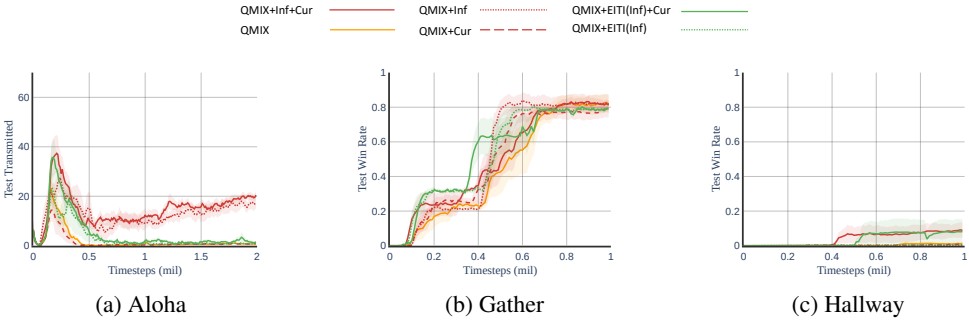

Figure 8: Performance of different kinds of exploration strategies on MACO benchmark.

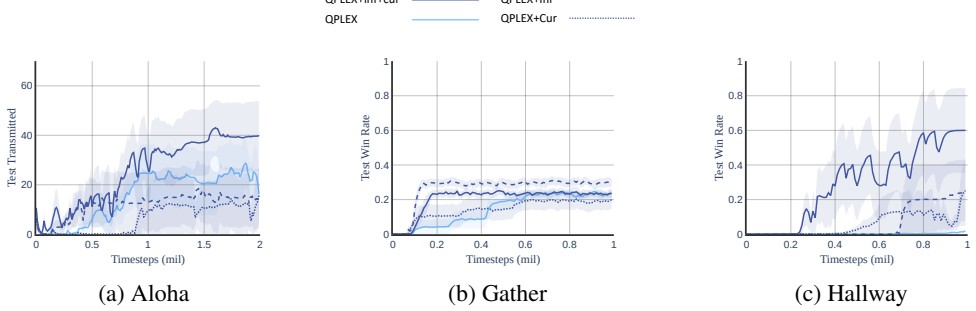

Figure 9: Ablations of our method when applying COIN to QPLEX.

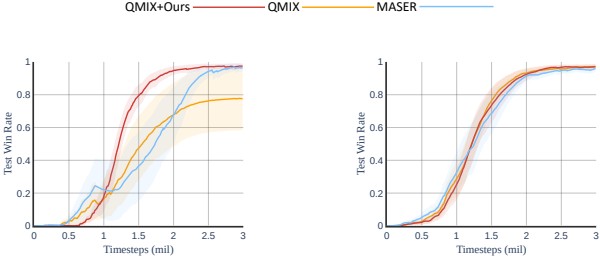

Figure 10: Results on StarCraft II sparse setting with map 3m(left) and 2s3z(right), where the rewards only come from death, killed or terminated.

