# OpenReview forum: "Two Heads are Better Than One: A Simple Exploration Framework for Efficient Multi-Agent Reinforcement Learning"
_NeurIPS.cc/2023/Conference — NeurIPS 2023 poster_

### Official Review · Reviewer_DahA · 2023-07-05

**Soundness:** 3 good
**Presentation:** 2 fair
**Contribution:** 2 fair
**Rating:** 4
**Confidence:** 4

**Summary:**

This paper propose a novel and compute-efficient exploration method: COIN, which can incorporate Curiosity-based and Influence-based exploration. For influence-based exploration，COIN quantify how each agent’s action affect the other agents and use the influence degree as the intrinsic reward to promote the exploration. For curiosity-based exploration，COIN uses the prediction errors of local observations of each agent and the global states to measure the uncertainty of the model to the environment. The experiments on three benchmarks StarCraft II, MACO, and Google Football show the superiority and effectiveness of COIN.

**Strengths:**

(1) This work designs a MARL framework which combines two kinds of main exploration methods and achieve sufficient and efficient exploration without bringing many computational costs.

(2) The exploration method proposed in this paper can be easily applied to other multi-agent methods.

(3) The overall idea of the paper is clear and easy to understand.


**Weaknesses:**

(1) To be honest，it lacks a certain degree of novelty. As is well known, the methods used in this paper like Qmix, MI and Prediction Error are all previous work. We just do some combination work.

(2) It should explain how to extend the method in this paper to other MARL methods.

(3) The existing experimental results seem insufficient，mainly reflected in the following aspects:

Many result curves do not show the final convergence state, but rather the intermediate state;

Lack of quantitative forms of expression, such as reward values or case study.

(4) The main body of this paper is clear to understand, here is space for improvement. I defer some of my issues in the appendix to "Questions".


**Questions:**

(1) In Equ.(1) and (2), the trajectory τkt should include state and action. So what is the meaning of p({a_k^t|\tau}_k^t\ )?

(2) Influence-based intrinsic reward is higher. This indicates that the action is more encouraging. Thus, In Equ.(8), why is“-”instead of“+”？

(3) Comparing with Fig.2, In Equ.(7), why is the input different between them ?

(4) The description of hyperparameters in the experiment can not be found.

(5) In Fig.5(a), why is the performance of Qmix better than Qmix+Inf and Qmix+Cur?

(6) Is the exploration framework of this paper attached to other MARL algorithms? How well?


**Limitations:**

In section 6, the authors talk about some limitations about this paper. At present, the potential negative social impact of this work has not been identified.

---

> ### Author Rebuttal · Authors · 2023-08-09
>
> **Q1:** It lacks a certain degree of novelty, and just do some combination of previous work such as MI and prediction error.
>
> **A1:** Off course, MI and prediction error are commonly used in RL.
> **In this paper, our main motivation is to highlight that different kinds of exploration play a different role in different scenarios. And we provide a simple but efficient way to combine two kinds of exploration strategies.**
> Compared with the related works, our proposed MI is (a) more reasonable and (b) easier to compute.
> More details are left in Appendix.
>
> **Q2:** Is the exploration framework of this paper attached to other MARL algorithms? How well?
> **A2:** Our framework is suitable for most of the MARL algorithms. Our method does not need to modify the baseline model, but only set up additional modules for computing the intrinsic rewards. These modules can be trained simultaneously with the original model.
>
> **Q3:** Many result curves do not show the final convergence state.  Lack of quantitative forms of expression, such as reward values or case study.
>
> **A3:** All randoms seeds of all baselines converge at the end in this paper.
> (1) Different random seeds may induce different outcomes with a large gap. (2) We add the smooth weight while drawing the curve.
> These are probably the reasons that it ``looks not to converge''.
>
> **Q4:** what is the meaning of $p(a_k^t|{\tau}_k^t)$?
>
> **A4:** ${\tau}_k^t$ is defined as $({o}_k^0, {a}_k^0,...,{o}_k^t)$ in this paper (see Preliminaries) which do not contains the actions in the current time-step. Hence, $p(a_k^t|{\tau}_k^t)$ means the probability to take action $a_k^t$ when given the current observations and history observation-action pairs.
>
> **Q5:** In Equ.(8), why is“-”instead of“+”?
>
> **A5:** We subtract (“-”) the influence-based intrinsic reward from $Q_i$ and get a pessimistic \~$Q_i$ to encourage the agents to influence each other during training.
> Considering the simple case of $y=r+\gamma*Q'-(Q-r_{inf})$.
> Abstract $r_{inf}$ from $Q$ is equivalent to add $r_{inf}$ to $r$.
> We apply the influence-based intrinsic rewards before each $Q_{i}$ is sent to the central critic because it can be scaled along with the ${Q}_{i}$.
>
> ** Q6:**  In Equ.(7), why is the input different between them?
>
> **A6:** Thanks for pointing out that. The inputs are the same, and we will rectify these typos in the revision.
>
> **Q7:** In Fig.5(a), why is the performance of Qmix better than Qmix+Inf and Qmix+Cur?
>
> **A7:** Fig.5(a) is an interesting finding. After watching the replay, we can provide an intuitive explanation. In this GRF scenario, each agent is initialized at the same location on the map. Hence, they need to explore the whole map in acceptable training time steps. Meanwhile, influence-based exploration will not promote such behavior, but only focus on the interactions among agents. This leads to too many unnecessary passes and makes the performance even worse than $\epsilon$-greedy.
> The sharing parameter paradigm leads to the same behavior of agents at the beginning of the training procedure. However, curiosity-based exploration will not promote the agents to influence each other and can not bring diversity for agents, which makes QMIX+Cur get worse.

---

> > ### Comment · Reviewer_DahA · 2023-08-18
> >
> > In this round of feedback, the author provided detailed modifications and explanations for the first round of feedback. Especially, the meaning of relevant formulas has been supplemented in depth, which enables readers to quickly and accurately grasp the content of the article. Unfortunately, the experimental part is still worth discussing, although the author has explained convergence and some experimental results, the relevant explanations may not be sufficient. If random seeds have a significant impact on experimental results, can it indirectly indicate that the stability of our method is weak? It is strongly recommended to refine and improve the experimental section. Finally, I hope the author can improve the relevant parts as soon as possible to make the article more competitive. Unfortunately, I am unable to modify the relevant scores.

---

> > > ### Author Response · Authors · 2023-08-18
> > >
> > > Thanks for recognizing our work. We appreciate the concerns raised about the experimental part, and here are the replies.
> > > 1. Indeed, random seeds can significantly influence the performance of deep reinforcement learning, regardless of training algorithms and exploration strategies [1-4]. Therefore, we followed recent studies and evaluated different methods using the mean or median of 10 random seeds. It is normal for a method to have high variance in performance in different seeds. For example, the baseline QMIX[1] in the scenario "3s5z_vs_3s6z" may have 0 and 100 win rate at two different seeds.
> > > 2. Moreover, the instability in MARL often refers to the fluctuations in performance caused by the large joint action and state space within one seed.
> > >
> > > Hence, the results cannot indicate that the stability of our method is weak. We will refine the experimental section to provide more clarity on this aspect.
> > > Thank you again for your feedback and we will work to improve the quality of our work.
> > >
> > > [1] Monotonic value function factorisation for deep multi-agent reinforcement learning
> > > [2] Weighted qmix: Expanding monotonic value function factorisation for deep multi-agent reinforcement learning.
> > > [3] Qplex: Duplex dueling multi-agent q-learning
> > > [4] Maser: Multi-agent reinforcement learning with subgoals generated from experience replay buffer

---

> ### Author Response · Authors · 2023-08-17
>
> Thanks for your careful review again.The discussion deadline is coming. If you have any other concern or question, welcome to discuss with us.

---

### Official Review · Reviewer_dhms · 2023-07-05

**Soundness:** 3 good
**Presentation:** 3 good
**Contribution:** 3 good
**Rating:** 6
**Confidence:** 3

**Summary:**

This paper investigates exploration methods in multi-agent reinforcement learning (MARL). Specifically, among the two main families of exploration methods, namely curiosity-based methods and influence-based methods, the paper reports that they could serve complementary roles to each other. Subsequently, the paper proposes a method that takes advantage of both types of exploration methods, demonstrating its effectiveness in 3 common MARL benchmarks.

**Strengths:**

1.	MARL exploration is an open and important problem for the field. The proposal of combining the two families of exploration methods in MARL is novel and the highlight of their complementary role is significant.
2.	The experimental results overall show the effectiveness of the proposed method against strong baselines
3.	The paper is well-written with the method well-explained.


**Weaknesses:**

1.	There are some inconsistencies in results across base algorithms of the proposed methods in different environments. Some additional ablation results could improve this (will be mentioned in the question section)
2.	The mentioned inconsistencies are not well explained, making the results less convincing.


**Questions:**

1.	The results of QMIX + ours in Aloha seem to be quite odd given the early burst and drop with the plateaued bad results. Any intuition or explanation for the phenomenon?
2.	Can you elaborate more on the possible reasons behind the performance discrepancies between the results of QMIX and QPLEX when adding your method?
3.	Do you have the results for QPLEX+ours in GRF? This could strengthen the results
4.	Given the drastic difference between QMIX and QPLEX when combined with the proposed exploration method, do the ablation results apply to QPLEX too
5.	Can you elaborate on how causal relationships could help in balancing the two terms?
6.	More of an open question, Could this have any failure cases specific to the multi-agent setting like the noisy TV issues in the single-agent case?
7.	Would these results apply in fully decentralized MARL methods without sharing parameters?


**Limitations:**

One limitation is discussed. No negative societal impact was discussed.

---

> ### Author Rebuttal · Authors · 2023-08-09
>
> **Q1:** The results of QMIX + ours in Aloha seem to be quite odd given the early burst and drop with the plateaued bad results. Any intuition or explanation for the phenomenon?
>
> **A1:** In Aloha, messages collided when two adjacent agents send messages simultaneously.
> Each successful transmission will gain a positive global reward (i.e., 0.1) and the collision will gain a negative global reward (i.e., -10).
> Meanwhile, the agents can not communicate with each other and do not know whether the action will cause a collision.
> In the beginning, they try to explore and hence get more successful transmissions. However, they gain more punishment at the same time.
> The severe punishment mechanism makes the agents adopt conservative strategies.
> That is the reason why the curve looks like this.
>
> **Q2:**  Can you elaborate more on the possible reasons behind the performance discrepancies between the results of QMIX and QPLEX when adding your method?
>
> **A2:** In most scenarios, COIN can improve the performance of both QMIX and QPLEX.
> The discrepancies between the results may come from environmental uncertainty and the method itself.
> The MARL scenarios are much more complicated than single-agent RL, thus, it is hard to say which method will definitely outperform the other.
> Even the same method with different random seeds can get different performance with a huge gap.
> Hence, it is a normal phenomenon where discrepancies exist, but it is hard for us to provide specific reasons for them.
>
> *Q3:** Do you have the results for QPLEX+ours in GRF? Do the ablation results apply to QPLEX too?
>
> *A3:** Our method improves QPLEX in GRF scenario, and the results are shown in Fig. (2) of the the affiliated file.
> We show the ablations of our method when apply on QPLEX in the affiliated file Fig. (1).
> Our method can improve QPLEX in different content in different scenarios except for a failure case ''Gather''.
>
> *Q4:** Can you elaborate on how causal relationships could help in balancing the two terms?
> *A4:** Given the causal relationships, we can know the potential interdependence among agents.
> We can assign lower $\beta_{inf}$ to the agents that do not rely on the others and higher $\beta_{inf}$ to the agents that most agents rely on. Meanwhile, we can simplify the computation of Eq.(3) to avoid computing the unnecessary MI.
>
> **Q5:** Could this have any failure cases specific to the multi-agent setting like the noisy TV issues in the single-agent case?
>
> **A5:** There is a failure case like ``noisy TV'' in the Gather scenario.
>
> A higher global reward of 10 is received if all agents choose the goal set at the beginning of the episode,
> and a relatively lower reward of 5 is received if no agents choose this goal.
>
> Obviously, QPLEX fails and falls into suboptimal results and our framework can not bring too much improvement.
>
> **Q6:** Would these results apply in fully decentralized MARL methods without sharing parameters?
>
> The experiments can also be applied in methods without sharing parameters, and the results could be even better.
> In some scenarios, the sharing parameters paradigm brings challenges and is harmful to MARL training. For example, in GRF, the sharing parameter paradigm leads to the same behavior of all agents at the beginning of the training procedure and hampers the possibility of the agents' cooperation.

---

> > ### Comment · Reviewer_dhms · 2023-08-12
> >
> > Thank you for your rebuttal. My concerns have been addressed. I Have raised my score to 6. I think the consideration of different types of exploration is useful. Even though the combination of methods seems simple, the perspective it brings has novelty.
> >
> > Please incorporate your discussions of the results in the rebuttal in the paper.

---

> > > ### Author Response · Authors · 2023-08-13
> > >
> > > Thanks for rasing the score! We will incorporate the discussions into the paper. If you have any other question or concern, welcome to discuss with us.

---

### Official Review · Reviewer_a6y3 · 2023-07-06

**Soundness:** 2 fair
**Presentation:** 3 good
**Contribution:** 2 fair
**Rating:** 6
**Confidence:** 4

**Summary:**

In this paper the authors propose a new framework to improve exploration in MARL. The method COIN improves exploration by combining the concepts in the literature of curiosity and influence.

**Strengths:**

This paper proposes a method that combines in a framework two popular concepts in the reinforcement learning literature, curiosity and influence, aiming to improve exploration. The authors analyse how the different components of their method affects their approach. A detailed description of how the method works is also provided and it is then tested in a set of different environments.

**Weaknesses:**

- "Li et al. [12] point that shared parameters among agents induce similar behaviors of players which makes the model fail to learn successful policies on this challenging task. Our influence-based intrinsic reward promotes the agents to affect each other and hence brings diversity." - this is a problem caused by the famous parameter sharing paradigm, that the authors also adopt in COIN; i feel that this is a big claim that I believes it needs further evidence.

Minor:
- line 156: "courage"->"encourage"
- line 96-97: "it shows unstable and innefective" poorly phrased
- l 144: "varriance" -> variance
- l 154: shouldnt it also be $\tau^~t$?
- line 283: plays -> play
- description of the dec pomdp in section 3 needs to be reviewed: for example, the observation model is missing and $\mathcal{O}$ represents the observation function; $\mathcal{N}$ inside the tuple G should be the number of agents and not the set; among others

Please find my questions below.

**Questions:**

1. In line 46 the authors state that their method is compute-efficient. Is there any evidence or proof for this?
2. in equation 7, does it mean that the only difference to predict state and observations is that to predict observations the agents use only the trajectories from a single time step? If so, is this accurate if we dont consider the previous values, since MARL is a non stationary sequential problem?
3. I have some questions about the influence of agents on others: does influence-based exploration mean that the agents are going to explore states where each agent has more influence on the actions of the others? If this is the case, how do they measure this influence? From my understanding, this is measured by summing the mutual information between all trajectories and the action of each agent individually to see the influence of each agent on the others; but does this guarantees that influence on the others is always good influence? i.e., can there be cases where we have bad influence and so these states shouldn't be explored?
4. In line 156: "Curiosity-based intrinsic rewards aim to courage the unfamiliar state, and influence-based intrinsic rewards aim to promote coordination." While curiosity based methods clearly directly tackle exploration problems, can we say the same from influence-based methods? Following the same logic, we could also say that almost any method in MARL is a method that tackles specifically exploration problems since most of them promote coordination.
5. the authors start the paper by mentioning that exploration is important in sparse reward environments; in this sense, have they tried to run their method on starcraft with sparse rewards too? Why choosing only normal stracraft environments in this case?
6. in section 5.3 it is unclear what the authors end up picking for $\beta_{inf}$ and $\beta_{cur}$: do they get the same weight, i.e., the same value?
7. there are also other methods that work towards efficient exploration, such as []. These methods, for example, use sub-goals instead of curiosity, which is also a common benchmark for some exploration-based methods. Have the authors considered comparing against some of these too?

[1] https://arxiv.org/abs/2206.10607

**Limitations:**

Please see above.

---

> ### Author Rebuttal · Authors · 2023-08-09
>
> **Q1:** ''Our influence-based intrinsic reward promotes the agents to affect each other and hence brings diversity." needs further evidence.
>
> **A1:** The parameter-sharing paradigm leads to the same behavior of different agents in Google Research Football(GRF). After watching the replay, we find that for the models without adding the influence-based intrinsic reward, different agents tend to perform the same action (same as Li et al.[1]). Meanwhile, influence-based rewards encourage each agent to influence the others' actions in the future, and hence can bring diversity.
>
> **Q2:** Is there any evidence or proof for compute-efficient?
>
> **A2:** COIN is compute-efficient for two main reasons:
> (1) First, we adopt simple formulations of both MI for influence-based intrinsic rewards and prediction errors for curiosity-based intrinsic rewards.
> (2) Both the MI and prediction errors can be estimated by neural networks, and the summation symbol in Eq.(5) and Eq.(6) can be computed in a single forward propagation.
> We will provide more discussions and emphasize them in the revision.
>
> **Q3:** Is the influence are measured by summing the mutual information?  Does this  that influence on the others is always good influence?
>
> **A3:** Yes, we define the influence of agent A on agent B as the MI, which means how agent A's current action will influence the future trajectory of agent B. And the summation of all the MI of agent A to its peers becomes the influence-based intrinsic reward of agent A.
>
> Our method can not guarantee the influence is always useful. Indeed, no method can guarantee the influence among agents is always good.
> Recent literature deems that promoting the interaction is often more useful to find meaningful states than exploring blindly.
> Meanwhile, what we want to highlight is that ''two heads are better than one'' which encourages the readers to try different kinds of strategies simultaneously rather than focusing on one single strategy.
>
> **Q4:** Curiosity based methods clearly directly tackle exploration problems. Following the same logic, we could also say that almost any method in MARL is a method that tackles specifically exploration problems since most of them promote coordination.
>
> **A4:** Curiosity-based methods encourage the agents to experience the unseen joint states.
> However, in the MARL setting, meaningful states are too sparse and hard to find.
> Therefore, how to find potentially meaningful states within acceptable explore time steps is a problem.
> Different kinds of exploration strategies follow different assumptions, and the influence-based methods assume that the states where the agents interact with each other are more likely to be meaningful.
> We find that the MARL scenarios are often complicated and only relying on one kind of strategy may not get the desired effect.
>
> **Q5:** Have tried to run on starcraft with sparse rewards ? Have the authors considered comparing against some methods with subgoals[1]?
>
> We perform experiments on sparse rewards, and compare our method with MASER[1], which is a goal-oriented method.
> The results are left in Fig. (3) of the affiliated PDF .
> Fig. (3) depicts that our method outperform MASER in 3m and 2s3z.
> Our results are slightly inconsistent with those reported in [1]. The reasons may come from the game version, random seed, et al.
> And we will perform more experiments to justity the superiority of our method.
>
> [1] https://arxiv.org/abs/2206.10607
>
> **Q6:** It is unclear what the authors end up picking for the hyperparameters?
>
> **A6:**  We mainly follow two principles to design the hyper-parameters: 1) The two kinds of intrinsic rewards are on the same order of magnitude. 2) The two scaled intrinsic rewards are about 100 to 1000 times lower than the maximum extrinsic reward at the beginning of the training.
> We choose different $\beta_{inf}$ and $\beta_{cur}$ for different scenarios which are shown in Appendix.

---

> > ### Comment · Reviewer_a6y3 · 2023-08-17
> >
> > Thanks for answering my questions. Regarding Q1, I am still not fully convinced that the proposed method will bring that much diversity. Watching replays might not be enough, as it might depend on the seeds, for example. Some sort of metric to evaluate this would be useful, or even looking at the weights of the trained networks could give a better idea of this diversity.
> >
> > The rest of my questions have been addressed. I believe the experiments in sparse settings and comparisons with MASER [1] are particularly important. I recommend incorporating the discussions and results of the rebuttal in the paper. I will raise my score, assuming that these will be included.

---

> ### Author Response · Authors · 2023-08-17
>
> Thanks for your careful review again.The discussion deadline is coming. If you have any other concern or question, welcome to discuss with us.

---

### Official Review · Reviewer_PR9o · 2023-07-21

**Soundness:** 2 fair
**Presentation:** 1 poor
**Contribution:** 2 fair
**Rating:** 6
**Confidence:** 4

**Summary:**

This paper looks at two different exploration strategies within Multiagent Reinforcement Learning. The two exploration strategies are curiosity-based and influenced based, where the latter is unique to MARL and takes into account the mutual information of one agents actions on the trajectory of another agent. The study then looks at combining these two forms of exploration and studies the effects of the individual intrinsic rewards as well as their combined effects on three distinct environments. A number of ablations are performed during these studies.

**Strengths:**

The paper introduces a novel algorithm which is able to explore in a MARL setting using two different additional intrinsic reward signals. It seems that such an exploration strategy is effective in the examples given at improving performance. The algorithm is relatively clean and the explanation is clear and precise. Exploration strategies are clearly a very important topic within the realm of computationally costly settings and so such an improvement is clearly important.

**Weaknesses:**

The language in the paper needs some significant work. I would recommend passing the paper through an LLM for checking for spelling and grammar throughout, as there are a fair number of such issues.

The lack of clear exploration in the hyperparameter space is a certainly an important limitation of this work. While the choice of hyperparameters is motivated, it is not clear that this has actually been studied, and in particular where the mixture of two intrinsic rewards is being used together, it seems that such an investigation would be very important to have a deeper understanding of their effects. Although an ablation study has been performed, this is really a binary investigation, whereas the continuum of mixings between the rewards would be most interesting.

Although there are a fair number of plots showing the learning trajectory on the test score, no summary statistics have been included. It would be useful to have a more detailed, quantitative set of measures of improvement, in particular over the hyperparameter sweeps.

**Questions:**

I believe that the weaknesses highlight the questions that I have, in particular surrounding the hyperparameters and a more detailed investigation of the statistics.

**Limitations:**

While limitations around the hyperparameters have been noted, these do not give sufficient response to the points raised above.

---

> ### Author Rebuttal · Authors · 2023-08-09
>
> **Q1:** The lack of clear exploration in the hyperparameter space. The continuum of mixings between the rewards would be most interesting. Quantitative set of measures of the hyperparameter sweeps.
>
> **A1:** The importance of two kinds of exploration is various in different time steps. Thus, it is very hard to decide which kind of strategy will dominate.
> In this work, We follow two principles to design the hyper-parameters: 1). Both two kinds of intrinsic rewards are on the same order of magnitude. 2). The two scaled intrinsic rewards are about 100 to 1000 times lower than the maximum extrinsic reward at the beginning of the training stage.
> We choose different $\beta_{inf}$ and $\beta_{cur}$ for different scenarios, which are shown in Appendix.
>
> Furthermore, it is worth noting that different scenarios require different exploration strategies, when it is hard to decide which one to use, implementing both in a simple and efficient way be a good choice.
>
> **Q2:** Checking for spelling and grammar throughout.
>
> **A2:** Thanks for your careful reviews and suggestions again. We will carefully proofread our paper and revise all typos.

---

> > ### Comment · Reviewer_PR9o · 2023-08-13
> > **Response to rebuttal**
> >
> > Thank you for the clarifications. I have now increased my score to a 6.

---

> > > ### Author Response · Authors · 2023-08-17
> > >
> > > Thanks for raising the score! If you have any other question or concern, welcome to discuss with us.

---

### Author Rebuttal · Authors · 2023-08-09

Thanks for your careful reviews and discerning comments. We will explain all the mentioned questions in the following, and some new experiments are added to the affiliated file.

---

### Decision · Program_Chairs · 2023-09-21

**Decision:**

Accept (poster)

**Comment:**

Its insights are not too surprising, and the algorithmic ingredients are not novel, but the results are useful on their own terms, specifically in how they demonstrate the benefits of qualitatively different exploration types in different circumstances. Also, while “pure” method papers tend to be easier to review, I find this to be an empirically well-executed hybridisation work and hence give it the benefit of the doubt.